# WebDevJudge: Evaluating (M)LLMs as Critiques for Web Development Quality

**Chunyang Li**[1,2]*, **Yilun Zheng**[1,3]*, **Xinting Huang**[1], **Tianqing Fang**[1], **Jiahao Xu**[3],
**Lihui Chen**[3], **Yangqiu Song**[2], **Han Hu**[1]

[1]Tencent AI Lab, [2]The Hong Kong University of Science and Technology,
[3]Nanyang Technological University
cliei@connect.ust.hk, yilun001@e.ntu.edu.sg

## Abstract

The paradigm of LLM-as-a-judge is emerging as a scalable and efficient alternative to human evaluation, demonstrating strong performance on well-defined tasks. However, its reliability in open-ended tasks with dynamic environments and complex interactions remains unexplored. To bridge the gap, we introduce **WebDevJudge**, a systematic benchmark for assessing LLM-as-a-judge performance in web development, with support for both non-interactive evaluation based on static observations and continuous interactive evaluation with a dynamic web environment. WebDevJudge comprises human preference labels over paired web implementations, annotated with structured and query-grounded rubrics to ensure high-quality ground truth. Using this benchmark, we comprehensively evaluate various evaluators, including LLMs, MLLMs, and agentic workflows. We systematically investigate the impact of different paradigms and guidance mechanisms. Our experiments reveal a significant gap between LLM judges and human experts. In-depth analysis indicates this gap stems from fundamental model limitations, including failures in recognizing functional equivalence, verifying task feasibility, and mitigating bias. Overall, WebDevJudge presents a challenge to LLM-as-a-judge, offering insights to guide future research toward developing more reliable and capable automated evaluators for complicated scenarios. Code and data are available at https://github.com/lcy2723/WebDevJudge.

## 1 Introduction

*Evaluate, refine, then evaluate again and refine again*—large language models (LLMs) have achieved remarkable success across various domains through this iterative cycle (Madaan et al., 2023; Chen et al., 2023; Shafayat et al., 2025). Conventional evaluation paradigms rely heavily on human assessment (Zhong et al., 2024; Starace et al., 2025), which, while meticulous, presents a critical bottleneck due to its high cost and low scalability. In response, the paradigm of LLM-as-a-judge has emerged as a promising alternative (Zheng et al., 2023; Dubois et al., 2024), offering a scalable and cost-effective solution for crucial development stages like verification (Lightman et al., 2024) and reward modeling (Yuan et al., 2025). With the advent of sophisticated language agents capable of planning, tool use, and collaboration (Yao et al., 2023; Qin et al., 2024a; Liang et al., 2024), the role of LLM-as-a-judge is rapidly expanding beyond basic, well-defined tasks to encompass challenging real-world problems (Zhuge et al., 2025; Bian et al., 2025). This progression is critical, as it paves the way for more capable automated evaluators, creating the possibility for language models to self-evolve in complex, real-world applications.

However, a fundamental question regarding the reliability of LLM-as-a-judge persists. Its effectiveness is well-established for static and basic tasks (Chen et al., 2024; Saha et al., 2025; Gou et al., 2025; Lù et al., 2025), but these successes share a critical commonality: they rely on static assessment of final outcomes. The reliability of LLM-as-a-judge in dynamic, open-ended domains

---

*Equal contributions. Work done by CL and YZ during their internship at Tencent.

involving complex interaction remains largely unexplored. Such contexts introduce significant challenges: dynamic evaluation requires continuous interaction with and comprehension of a changing environment (Li et al., 2023a; Paglieri et al., 2025), while their open-ended nature necessitates establishing feasible assessment standards (Li et al., 2025). This gap between the expanding scope of automated judges and the lack of rigorous validation in complex, interactive settings highlights an urgent need for a new meta-evaluation benchmark. Such a benchmark is essential to assess, interpret, and ultimately enhance the reliability of LLM-based judges as we approach increasingly autonomous AI systems.

Consequently, in this work, we introduce **WEBDEVJUDGE**, a meta-evaluation benchmark for assessing LLM-as-a-judge on a representative complex and interactive task using the context of web development. Web development offers an ideal testbed for complex dynamic evaluation, as it inherently has interaction requirements, whose assessment depends not just on static code but on real-time interaction. The task is also intrinsically open-ended, lacking a single absolute answer. To establish a high-quality ground truth, we introduce a structured annotation methodology using query-grounded *rubric trees*, which decomposes high-level requirements into a verifiable hierarchy of fine-grained criteria. This rigorous protocol combining rubric with human judges yields an inter-annotator agreement over 80%, which substantially exceeds the 63% reported for MT-Bench (Zheng et al., 2023), confirming the reliability of the preference labels. Departing from traditional benchmarks focused on static text (Tan et al., 2025), WEBDEVJUDGE supports both static and interactive evaluation by offering multifaceted representations of each web implementation, including its source code, screenshot of the rendered webpage, and a fully interactive environment for dynamic assessment, as shown in the right part of Figure 1. Evaluator performance is measured by agreement with expert human preferences on paired web implementations—a methodology widely adopted for nuanced, open-ended tasks where the absolute answer is deficient (Bai et al., 2022; Lambert et al., 2025).

Using WEBDEVJUDGE, we conduct a comprehensive evaluation of a wide array of judges, including LLMs, MLLMs, and agentic workflows, under various paradigms and guidance mechanisms. Our experiments reveal that a significant capabilities gap persists between the advanced models and human experts, with a performance discrepancy of about 15%. Notably, different guidance strategies provide only marginal improvements in the pairwise comparison setting, suggesting that preference prediction through comparative assessment represents an internalized capability in models (Hua et al., 2025; Yu et al., 2025). Furthermore, while agentic workflows appear well-suited for interactive task evaluation, they fail to outperform vanilla models due to error accumulation across planning and execution stages (Pan et al., 2025). Through detailed error analysis and case studies, we systematically investigate the failure modes of automated evaluators. As part of this analysis, we construct **WebDevJudge-Unit**, a diagnostic dataset specifically designed to evaluate feasibility verification capabilities of different types of LLM-as-a-judge. Our investigation reveals two fundamental performance bottlenecks: (1) a persistent inability to recognize *functional equivalence* between diverse implementations that achieve the same objectives through different approaches or terminologies, such as implementations using the same title element with only variations in text, and (2) systematic weaknesses in *feasibility verification*, where static assessment suffers from low precision due to static code analysis limitations while interactive agents exhibit low recall stemming from their own operational constraints. These compounding errors ultimately undermine evaluator performance, pointing toward fundamental research directions for developing truly reliable automated evaluators in complex, interactive domains. Our main contributions are as follows:

- We construct WEBDEVJUDGE, a meta-evaluation benchmark that supports both static and interactive assessment of web development quality with high-quality preference labels.
- Comprehensive empirical evaluation of (M)LLMs and agentic workflow reveals that current LLM-as-a-judge still falls short of human-level reliability in preference prediction.
- Detailed error analysis identifies the systematic weaknesses of LLM-as-a-judge, providing critical insights for developing more reliable automated evaluators.

## 2 RELATED WORK

**LLM-as-a-Judge** The paradigm of LLM-as-a-Judge leverages powerful large language models to simulate human-like assessment, enabling scalable and cost-efficient evaluation (Gu et al., 2025). This approach has been widely adopted across various domains, including question answering (Bai

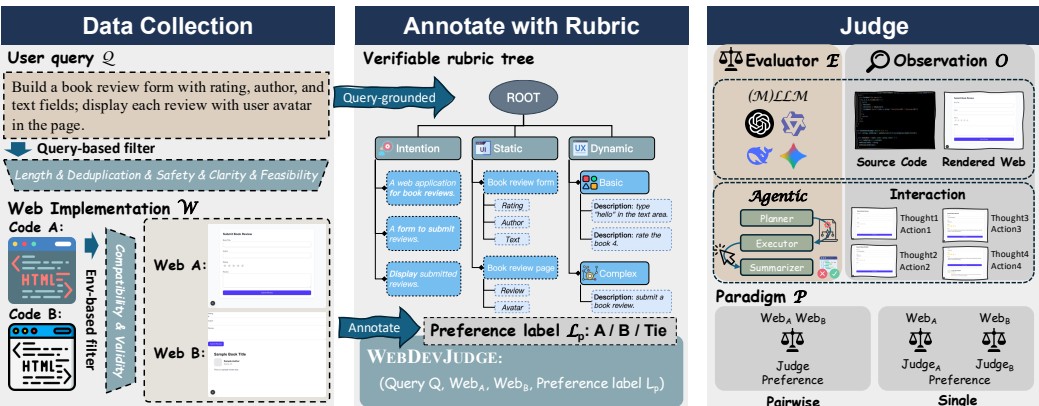

Figure 1: Overview of WEBDEVJUDGE. *Left*: Data Collection with query-based and environment-based filtering. *Center*: Preference label annotation with verifiable rubric tree. *Right*: Evaluate (M)LLM-based and agentic evaluators under pairwise and single-answer paradigms.

et al., 2023), data filtering (Li et al., 2024; Xu et al., 2025), and trajectory evaluation (Pan et al., 2024; Xue et al., 2025). Typical implementations employ a strong LLM as an evaluator, which compares or scores candidate responses against predefined criteria. Common comparison methods include pairwise comparison and single-answer grading Zheng et al. (2023). As application scenarios broaden, the conventional LLM-as-a-Judge approach is evolving into Agent-as-a-Judge (Zhuge et al., 2025), wherein LLM-based agents are equipped with tool-using and collaboration capabilities. Gou et al. (2025) propose a judge agent with extractor and verifier to evaluate deep research tasks with rubric trees, while Zhuge et al. (2025) design a modular agentic framework to evaluate agentic systems. Despite its efficiency, this method faces several challenges. Position bias (Wang et al., 2024) and verbosity (Ye et al., 2025) preference may skew results, and the creation of detailed, human-written rubrics is both labor-intensive and difficult to scale (Starace et al., 2025). Additionally, typical assessments may overemphasize final outcomes, limiting their applicability in open-ended interactive tasks (Zhang et al., 2025a). To evaluate the judge capability of LLMs, we introduce a benchmark on open-ended web development scenarios. It is designed to assess a wide range of LLM-based evaluators, with the dual purpose of benchmarking their performance and exposing fundamental shortcomings in existing automated evaluation approaches.

**Meta Evaluation**   Meta-evaluation accesses the effectiveness of automated evaluation by measuring its correlation with human or ground truth. Existing benchmarks primarily focus on two aspects: (1) alignment with preference label, often using pairwise comparisons (Zheng et al., 2023; Li et al., 2023b), and (2) accuracy in identifying correct task outcomes, particularly in reasoning (Luo et al., 2023; He et al., 2025) and agentic tasks (Lù et al., 2025). Prior works such as MT-bench (Zheng et al., 2023) and LLMEval (Zhang et al., 2023) measure how LLM judges mirror human preferences in multi-turn conversational and instruction following tasks. However, these benchmarks may be constrained by human subjectivity and position bias (Wang et al., 2024). Alternatives like LLM-Bar (Zeng et al., 2024) and JudgeBench (Tan et al., 2025) test judges on re-annotated instruction adherence or verifiable reasoning discrimination. These benchmarks mainly focus on text-based tasks without complex environments. In the context of interaction, AgentRewardBench (Lù et al., 2025) uses expert-annotated result of pre-scripted trajectories to benchmark how well LLM evaluators score agent performance. ArtifactsBench (Zhang et al., 2025a) evaluates dynamic visual effects. However, it lacks a critical component of assessing environmental changes that are driven by real-time user input. Our work, WEBDEVJUDGE, introduces a meta-evaluation benchmark that assesses judges on dynamic, real-world web development tasks, emphasizing continuous interaction with live web environments, offering insights in complex interactive settings.

## 3 WEBDEVJUDGE BENCHMARK

WEBDEVJUDGE serves as a meta-evaluation benchmark designed to assess whether LLM-as-a-judge can effectively approximate human preference judgments in web development tasks. We frame the task as a preference evaluation problem. Each instance is represented as a quadruple $(Q, W_a, W_b, l_p)$, where $Q$ denotes a web development query (e.g., "build a book review page"), $W_a$ and $W_b$ represent web implementations from two distinct models $a$ and $b$, and $l_p$ is the annotated preference label between the two outputs. The objective is to predict the preference label—i.e., whether $a$ wins, $b$ wins, or the result is a tie—which is widely used for open-ended tasks (Zeng et al., 2024; Lambert et al., 2025). The observations of $W_a$ and $W_b$ may take various forms across experimental settings, such as code, screenshots, or interaction trajectories.

### 3.1 DATA FILTERING

We collect data from the `webdev-arena-preference-10k` dataset (Vichare et al., 2025; Chiang et al., 2024), which comprises 10,501 user queries with paired code outputs from 2 models and user-provided preferences. We apply a two-stage filtering process to enhance data quality:

**Query-based filtering** We first exclude extremely short queries and verbatim duplicates. The remaining queries are further filtered via an LLM according to the following criteria: (1) *Safety*: exclusion of harmful or offensive content; (2) *Clarity*: the query must be unambiguous and interpretable; (3) *Feasibility*: the query should be realistically implementable as a web application, based on its purpose and required level of interaction.

**Environment-based filtering** We deploy each web implementation in a unified execution environment. Instances that fail to deploy correctly or require niche dependencies are discarded. To ensure validity, we capture initial screenshots of each rendered webpage and use a multi-modal LLM to filter out invalid cases (e.g., blank pages or intrinsic errors).

After applying these filters, we retain 1,713 high-quality instances. We sample 700 instances for further annotation. Details regarding the data construction process are provided in Appendix B.

Table 1: Annotation agreement rates with and without the verifiable rubric. The 'without rubric' part shows agreements between: (1) annotators and (2) annotators and the original labels. The 'with rubric' part shows inter-annotator agreements under human-written and LLM-generated rubrics.

| Setting | Without rubric | | With rubric | |
|---|---|---|---|---|
| | *inter-annotator* | *annotator-origin* | *Human-written* | *LLM-generated* |
| `w/ tie` | 65.0 | 53.0 | 92.0 | 90.0 |
| `w/o tie` | 91.3 | 77.4 | 95.5 | 95.1 |

### 3.2 ANNOTATION VIA RUBRIC TREE

Consistent with prior work (Zheng et al., 2023; Tan et al., 2025), we note that raw preference labels may reflect subjective bias rather than objective quality. To quantify this issue, we sample 100 instances and have them re-annotated by two expert annotators based on fully deployed web instances. As shown in Table 1, both inter-annotator agreement and agreement with original labels are initially low, highlighting the need for a more structured and standard annotation protocol.

To address this, we introduce rubric tree—a structured evaluation framework commonly used in complex assessment scenarios (Starace et al., 2025; Gou et al., 2025). The tree is query-based and scalable by design, organized along three core dimensions: intention, static quality, and dynamic behavior (see Figure 1). Each leaf node corresponds to a binary test, whose outcomes are aggregated hierarchically to parent nodes. This allows for both a holistic score at the root and fine-grained diagnostic insight via leaf-level pass rates.

We validate the effectiveness of the rubric tree by manually constructing trees for 50 instances and asking two annotators to evaluate these using the rubric as a reference. As shown in Table 1, rubric-

guided annotation significantly improves agreement rates. To scale this process, we employ few-shot LLM generation to automatically produce rubric trees. A third annotator labels the same instances using these generated rubrics, achieving high agreement with human-written rubrics—confirming the utility of LLM-generated rubrics. We then use generated rubric trees to annotate the remaining data via two expert annotators with software engineering backgrounds. During annotation, we perform manual inspections to exclude incompatible and harmful cases. Final inter-annotator agreement on a sampled subset reaches 89.7%, demonstrating high consistency.

Table 2: Categories and their respective sub-categories of queries in WEBDEVJUDGE.

| Category | Sub-categories | Total % |
|---|---|---|
| **DIGITAL DESIGN** | Website Design
UI Design | 34.0 |
| **GAME AND APP DEVELOPMENT** | Game Development
App Development
App Design | 34.4 |
| **WEB AND SPECIALIZED TECHNOLOGIES** | Clone Development
Web Development
Simulations
AI Applications
Multilingual Queries
Digital Tools
Creative Humor | 31.6 |

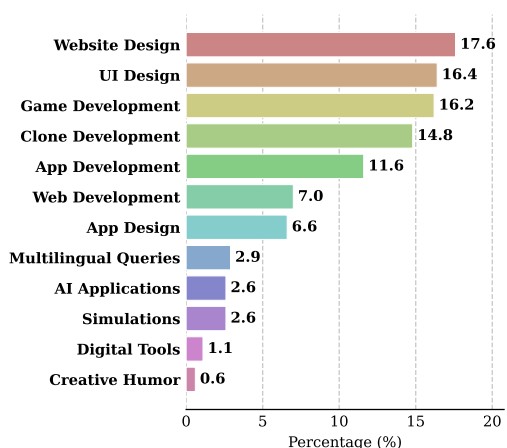

Figure 2: The distribution of sub-category across WEBDEVJUDGE.

## 3.3 DATA STATISTICS

The final benchmark consists of 654 instances, with preference distributions as follows: 269 for $a$, 276 for $b$, and 109 ties. To characterize the query domain coverage, we conduct a topic analysis based on the original topics of the `webdev-arena-preference-10k` dataset (Vichare et al., 2025). Fine-grained topics are grouped into three broad categories: Digital Design, Game and App Development, and Web and Specialized Technologies based on their shared characteristics and application contexts of representative examples. Table 2 and Figure 2 provide detailed category distributions and visualizations.

## 4 EXPERIMENTS

We conduct comprehensive experiments and evaluate different evaluation settings on WEBDEVJUDGE, focusing on the following research questions: (1) Whether LLM-as-a-judge can be an alternative to human preference in open-ended complex tasks like web development? (2) How do different settings and strategies affect the agreement rate performance?

### 4.1 EVALUATOR PERFORMANCE ON WEBDEVJUDGE

**Setup** Following (Zheng et al., 2023; Xie et al., 2025), we evaluate the evaluators under 2 distinct paradigms:

- *Pairwise comparison*: This approach directly compares two responses, leveraging relative judgments to infer preference, an intuitive and natural fit for preference prediction.
- *Single answer grading*: This approach employs LLMs to assign scores or labels to individual responses, with preferences derived by comparing scores or pass rates.

To enable structured and domain-grounded assessment, we design a multi-dimensional Likert scale inspired by prior work (Lan et al., 2024; Zhang et al., 2025a) and incorporate principles from international software testing standards (ISO/IEC/IEEE, 2022). Our criteria comprises four dimensions,

Table 3: Agreement Rate (%) of different evaluators under different evaluation paradigms. The best average performance of the whole dataset is highlighted in **bold** and the second best is underlined.

| Model/Method | DIGITAL DESIGN | | GAME & APP | | WEB & SPECIAL | | AVERAGE | |
|---|---|---|---|---|---|---|---|---|
| | *Single* | *Pair* | *Single* | *Pair* | *Single* | *Pair* | *Single* | *Pair* |
| **Vanilla** | | | | | | | | |
| **Non-reasoning Models** | | | | | | | | |
| 🖼 GPT-4.1 | 57.66 | 70.72 | 64.89 | 73.33 | 59.90 | 66.67 | 60.86 | **70.34** |
| 🖼 GPT-4o | 53.15 | 68.02 | 59.56 | 70.67 | 60.39 | 64.25 | 57.65 | 67.74 |
| 🖼 Qwen-2.5-VL-72B-Inst. | 46.85 | 68.02 | 51.11 | 66.22 | 52.17 | 64.73 | 50.00 | 66.36 |
| 🖼 Gemini-2.5-flash-lite | 50.45 | 63.51 | 51.56 | 60.89 | 48.31 | 63.77 | 50.15 | 62.69 |
| DeepSeek-V3-0324 | 60.36 | 67.12 | 59.11 | 66.67 | 60.39 | 67.63 | 59.94 | 67.13 |
| Kimi-K2-Inst. | 62.61 | 68.02 | 62.22 | 67.56 | 54.59 | 66.18 | 59.94 | 67.28 |
| Qwen3-235B-A22B-Inst. | 55.86 | 66.22 | 63.11 | 68.89 | 60.87 | 62.80 | 59.94 | 66.06 |
| Qwen3-30B-A3B-Inst. | 58.56 | 68.02 | 63.56 | 65.33 | 57.00 | 65.70 | 59.79 | 66.36 |
| Qwen2.5-32B-Inst. | 50.45 | 63.51 | 54.22 | 67.56 | 52.17 | 63.29 | 52.29 | 64.83 |
| Gemma-3-27B-it | 45.95 | 61.71 | 54.22 | 57.78 | 46.38 | 61.35 | 48.93 | 60.24 |
| Qwen2.5-14B-Inst. | 48.65 | 58.11 | 46.22 | 54.67 | 39.61 | 51.21 | 44.95 | 54.74 |
| **Reasoning Models** | | | | | | | | |
| 🖼 Claude-4-Sonnet | 54.05 | 72.52 | 63.11 | 72.00 | 60.39 | 65.70 | 59.17 | 70.18 |
| 🖼 Claude-3.7-Sonnet | 67.57 | 70.72 | 64.44 | 66.22 | 59.42 | 70.05 | **63.91** | 68.96 |
| 🖼 Gemini-2.5-pro | 51.80 | 64.41 | 53.33 | 72.00 | 45.41 | 60.39 | 50.31 | 65.75 |
| GLM-4.5 | 58.56 | 71.17 | 56.89 | 70.67 | 51.69 | 63.77 | 55.81 | 68.65 |
| DeepSeek-R1-0528 | 51.80 | 69.37 | 61.33 | 68.89 | 50.24 | 62.32 | 54.59 | 66.97 |
| Qwen3-30B-A3B-Think. | 53.15 | 64.86 | 55.56 | 64.00 | 53.14 | 62.80 | 53.98 | 63.91 |
| **Agentic** | | | | | | | | |
| Intention rubrics | 38.74 | - | 32.00 | - | 48.31 | - | 39.45 | - |
| Static rubrics | 52.25 | - | 48.00 | - | 53.14 | - | 51.07 | - |
| Dynamic rubrics | 54.05 | - | 59.11 | - | 53.62 | - | 55.66 | - |
| Combined rubrics | 59.91 | - | 58.22 | - | 63.77 | - | 60.55 | - |
| **Human** | | | | | | | | |
| Pairwise comparison | - | 84.23 | - | 83.11 | - | 86.47 | - | 84.56 |

Functionality, UI Quality, Code Quality, and Interactivity, each with several sub-criteria rated on a 5-point Likert scale (1: lowest, 5: highest). Detailed criteria are provided in Appendix C.

**Implementation**    We evaluate two types of evaluators: (1) **Vanilla (M)LLMs**: we test models from OpenAI (OpenAI, 2025; 2024), Anthropic (Anthropic, 2025b;a), Google (Team, 2025a), Qwen (Team, 2025e;d), DeepSeek (DeepSeek-AI, 2024; 2025), Moonshot (Team, 2025c), and ZAI (Team, 2025b). For *single answer grading*, we provide the query, code, and evaluation criteria; we further include an initial screenshot of the rendered webpage for MLLMs. The model is prompted to output scores for each sub-criterion. For *pairwise comparison*, we supply the query, code of $W_a$ and $W_b$, evaluation criteria, and screenshots of both rendered webpages (if applicable). The evaluator scores both $W_a$ and $W_b$ per sub-criterion, with explicit instructions to ignore position bias and remain objective. Preference is determined by comparing total scores. (2) **Agentic Workflow**: similar to Bian et al. (2025), we model evaluation as a multi-stage pipeline with a planner, an executor, and a summarizer:

$$\text{Query} \xrightarrow{\text{Planner}} \text{Plan with test cases} \xrightarrow{\text{Executor}} \text{Results} \xrightarrow{\text{Summarizer}} \text{Judge} \qquad (1)$$

The planner generates a verifiable evaluation plan conditioned on the query. The executor runs test cases, and the summarizer synthesizes results into a judgment. We use the generated rubric tree as the evaluation plan and reference for summarization. Preferences are inferred by comparing summarized outcomes. In implementation, we utilize UI-TARS-1.5 (Seed, 2025), one of the state-of-the-art GUI agents, as the executor, to ensure the reliability of the evaluation process.

**Result** The main experimental results are summarized in Table 3. Our analysis yields several key observations on the performance of LLM-based evaluators in web development tasks.

**LLM-as-a-judge falls short of human-level reliability on complex evaluation tasks.** The primary finding is that no current model achieves a sufficient level of agreement with expert judgments. The top-performing evaluator, GPT-4.1 under the pairwise paradigm, attains an agreement rate of 70.34%. This gap underscores the inherent complexity of web development evaluation, which demands a holistic assessment of functionality and aesthetics, especially for the "tie" cases. We also observe a clear performance ceiling: while smaller models exhibit scaling effects, larger and more capable models show diminishing returns, consistently plateauing around the 70% agreement rate.

**Pairwise comparison is a far more effective paradigm for preference evaluation.** Across the board, the pairwise paradigm yields an average improvement of over 8.0% in agreement rate compared to single-answer grading. Relative judgment helps models focus on discriminative features between two candidates, reducing the need for absolute quality calibration. In contrast, single-answer grading not only performs worse but also leads to inconsistent model rankings—some models deviate from their expected benchmark ordering (Zhang et al., 2025b), indicating that this paradigm is less suitable for open-ended, multi-faceted tasks. This may be due to the cognitive load of applying multi-point Likert scales consistently (Ouwehand et al., 2021), which demands a calibrated internal standard of quality that is difficult for both humans and LLMs to maintain. Simplified evaluation mechanisms, such as binary checklists, may be more effective; we explore this further in Section 4.2.

**Agentic workflow suffers from compounding errors.** We evaluated the agentic workflow using rubrics from different aspects, as well as a comprehensive score integrating all three. The results show that it performs significantly better on the Dynamic aspect, which involves strong interactive properties, compared to others with weaker interactivity. Counter-intuitively, the agentic workflow fails to outperform the vanilla models. This appears to result from error accumulation across its multi-stage process. We identify two primary failure modes:

- **Brittle Planning:** The *Planner* struggles with the ambiguity of user queries that are often not expressed with expert-level precision. This leads to the generation of evaluation plans that are either too generic to be discriminative or overly specific, causing failures due to minor implementation variations.

- **Faulty Execution:** The *Executor* agent's ability to navigate the web and verify task states remains unreliable. It may misinterpret outcomes, injecting noise into the evaluation process. This unreliability is investigated more deeply in Section 4.3.

Errors compound across the planner-executor-summarizer pipeline, reducing the reliability of the final judgment compared to an end-to-end evaluation by a single model (Chen et al., 2025b).

## 4.2 DIFFERENT EVALUATION GUIDES AND OBSERVATION FORMS

To further diagnose factors affecting evaluator performance, we conduct controlled experiments focusing on two key variables: the structure of evaluation guidance and the form of observation.

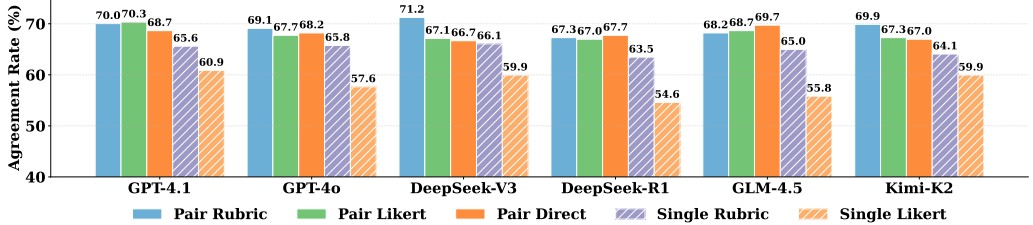

Figure 3: Agreement rates of LLM evaluators under various guidance protocols. We compare pairwise and single-answer paradigms using direct judgment, Likert scales, and structured rubrics.

**Evaluation Guidance**   We investigate how different evaluation protocols influence judgment accuracy, comparing three widely-used forms of guidance (Zheng et al., 2023; Gou et al., 2025): (1) *Direct*: The evaluator provides a preference without explicit criteria. (2) *Likert Scale*: As used in Section 4.1, the evaluator scores outputs along predefined dimensions using a multi-point scale. (3) *Rubric*: The evaluator assesses outputs using the rubric trees from our annotation process, with final scores computed as a weighted aggregation of binary leaf-node pass rates across three core dimensions (intention, static, and dynamic).

Results in Figure 3 reveal a noteworthy finding: under the pairwise evaluation, the *Direct* setting achieves agreement rates comparable to guidance-based methods. This phenomenon aligns with observations in instruction following tasks (Zeng et al., 2024), where models perform similarly on agreement with or without metric-based guidance. This result suggests that evaluation capability is an internalized skill in modern LLMs, and external guidance provides only marginal benefits in relative assessment settings (Qin et al., 2024b). Furthermore, we find that reasoning models exhibit better performance under the *Direct* condition, indicating that imposing rigid structured metrics might constrain the models' inherent reasoning processes, thereby limiting their evaluation potential. Furthermore, we validate the hypothesis from earlier: in single answer grading, the binary *Rubric* approach substantially outperforms the multi-point *Likert* scale, reinforcing that verifiable evaluation protocols yield more reliable judgments when there is a lack of relative information.

Table 4: Impact of observation forms on the performance of multimodal evaluators. The numbers in parentheses indicate the performance change relative to the setting with both code and image inputs.

| Model | Single Answer Grading | | | Pairwise Comparison | | |
|---|---|---|---|---|---|---|
| | Image Only | Code Only | Both | Image Only | Code Only | Both |
| Claude-4-sonnet | 55.66 (↓ 3.51) | 61.77 (↑ 2.60) | 59.17 | 59.48 (↓ 10.7) | 67.58 (↓ 2.60) | 70.18 |
| GPT-4.1 | 54.59 (↓ 6.27) | 62.69 (↑ 1.83) | 60.86 | 60.55 (↓ 9.79) | 69.57 (↓ 0.77) | 70.34 |
| GPT-4o | 53.36 (↓ 4.29) | 56.27 (↓ 1.38) | 57.65 | 59.79 (↓ 7.95) | 67.43 (↓ 0.31) | 67.74 |
| Gemini-2.5-pro | 53.36 (↑ 3.05) | 50.15 (↓ 0.16) | 50.31 | 62.08 (↓ 3.67) | 64.53 (↓ 1.22) | 65.75 |
| Qwen-2.5-VL-72B | 51.53 (↑ 1.53) | 50.92 (↑ 0.92) | 50.00 | 59.63 (↓ 6.73) | 64.68 (↓ 1.68) | 66.36 |

**Influence of Observation Modality**   We next analyze how the input modality affects multimodal evaluators, comparing three observation forms: (1) `Image Only`, providing only the initial screenshot of the rendered web page; (2) `Code Only`, providing only the source code; and (3) `Both`, providing both code and screenshot. As shown in Table 4, code emerges as the most critical modality for evaluating web development tasks. Withholding code leads to a significantly larger performance drop than withholding screenshots, suggesting that while MLLMs can process visual input, their judgments are fundamentally anchored in the structured source code (Wang et al., 2025). In *pairwise comparison*, using both modalities yields the best results, indicating that visual context provides complementary signals that aid in refining relative judgments (Chen et al., 2025a).

## 4.3   ERROR ANALYSIS

To systematically diagnose the failure modes of LLM-based evaluators, we conduct a fine-grained analysis along three key dimensions: inherent biases, limitations in understanding functional equivalence, and shortcomings in feasibility analysis.

Table 5: Positional bias in pairwise comparison. Preference for specific position, consistency and the absolute difference in agreement rate ($\Delta$ AR) between original and swapped orders are reported.

| Model | Consistency | | First | | Second | | $\Delta$ AR | |
|---|---|---|---|---|---|---|---|---|
| | Direct | Likert | Direct | Likert | Direct | Likert | Direct | Likert |
| Claude-4-sonnet | 89.6 | 87.9 | 5.7 | 3.7 | 4.7 | 8.4 | 0.61 | 1.53 |
| GPT-4.1 | 83.3 | 85.2 | 0.9 | 3.0 | 15.8 | 11.8 | 0.30 | 1.07 |
| DeepSeek-V3-0324 | 84.1 | 83.5 | 11.6 | 7.3 | 4.3 | 9.2 | 1.53 | 1.23 |

**Instruction**: Consider a requirement met if the solution's feature is equivalent. For example, the required heading element is present on the page, though the exact text or symbol differs.

**Requirement**:
A grading row for the "Demonstration" category
- A text label displaying "Demonstration".
- A set of star icons for rating "Demonstration".

**Element in webpage**:
Presentation          0 / 5
★ ★ ★ ★ ★

**LLM**: false          **Agent**: false          **Human**: true

Figure 4: An example illustrating the failure to recognize functional equivalence. The webpage element for "Presentation" serves the same purpose as the required "Demonstration" rating. Detailed examples can be found in Appendix F.2.

**Inherent Biases in Judgment** Prior studies (Ye et al., 2025) have revealed that LLM-as-a-judge exhibits various forms of bias, such as positional and verbosity biases. We focus specifically on positional bias. Despite explicit instructions to ignore order and remain objective, models still exhibit a systematic preference for responses in a specific position, as shown in Table 5. One might hypothesize that this bias emerges primarily in ambiguous cases where the two options are of comparable quality. However, our analysis of the label distribution for instances with inconsistent predictions reveals that the proportion of ties is not higher than that of wins or losses. This finding suggests that positional bias is not merely an artifact of ambiguity but rather an inherent deficiency in the models, and instruction alone is insufficient to eliminate these deeply embedded inductive biases. Rather than employing debiasing techniques such as swapping, we choose the prompting method to reflect the models' authentic single-pass evaluation capability. Any inherent bias (such as position bias) is considered an intrinsic flaw of the model's judgment ability. A further comparison with the debiasing technique is provided in Appendix E.1.

**Limitations in Understanding Functional Equivalence** A critical requirement for reliable evaluation is the ability to recognize functional equivalence—i.e., determining whether different implementations satisfy the same underlying requirement. We find that evaluators often fail in such judgments, adhering strictly to literal interpretation rather than intent. For example, as illustrated in Figure 4, when instructed explicitly, human evaluators correctly recognize an element implemented with alternative text, whereas both LLM and agentic evaluators incorrectly reject it. This reflects a fundamental gap in contextual and pragmatic reasoning, limiting their applicability in open-ended domains where diversity in implementation is common.

**Shortcomings in Feasibility Analysis** We further investigate the evaluators' ability to accurately verify task fulfillment. Existing benchmarks such as AgentRewardBench (Lù et al., 2025) are too general to assess web-specific judgment. To address this, we construct **WebDevJudge-Unit**, a targeted dataset of 502 test cases. Each instance consists of web code, a specific verification task, an expected result, and a label indicating whether the task is feasible. We evaluate both LLM-based (code-

Table 6: Performance on the *WebDevJudge-Unit* for feasibility verification. We report precision (P), recall (R), F1-score, and accuracy (Acc).

| Model | Observe | P | R | F1 | Acc |
|---|---|---|---|---|---|
| UI-TARS-1.5 | Traj. | 82.4 | 70.3 | 75.8 | 75.1 |
| GPT-4.1 | Code | 72.1 | 90.0 | 80.1 | 75.1 |
| DeepSeek-V3 | Code | 69.6 | 93.5 | 79.8 | 73.7 |

only) and agent-based (interaction-driven) evaluators on this dataset. From the results in Table 6, we identify complementary weaknesses: (1) For *LLM evaluators*, these models achieve high recall but suffer from low precision. While they can often identify relevant code segments, they are unable to verify actual execution outcomes, leading to false positives when the code appears relevant but does not correctly implement the desired functionality. (2) For *agentic evaluators*, they exhibit higher precision but lower recall. They are effective when they successfully execute a test plan, but sometimes fail to complete tasks due to limitations in navigation or state interpretation. Consequently, they incorrectly label feasible tasks as infeasible due to their own operational failures rather than actual shortcomings in the web implementation. This divergence highlights a fundamental trade-off: static code analysis lacks execution grounding, while interactive agents are constrained by their

own operational reliability. An ideal evaluator would combine the comprehensive coverage of code-aware reasoning with the grounded verification of interactive testing. To further investigate this, we design a gated model ensemble strategy to leverage these strengths in the single-comparison for rubric. Specifically, we adopted the results of the LLMs for the intention and static tasks. For the dynamic tasks, we implemented the following logical expression:

$$Res_{\text{dynamic}} = Agent \vee LLM \tag{2}$$

This formulation is motivated by the performance trade-off between the constituent components: the Agent provides high-precision filtering for dynamic elements, whereas LLMs ensure higher recall. This hybrid approach optimizes the overall detection capability by balancing exactness with coverage. We selected GPT-4.1, DeepSeek-V3-0324, and UI-TARS-1.5 for this ensemble. The results are shown in Table 7. These results demonstrate that combining static LLM with agent-based dynamic verification can enhance the overall evaluation agreement.

Our analysis reveals that the limitation of LLM-as-a-judge lies in fundamental deficiencies in calibration capability. Lacking this calibration, LLMs struggle to map abstract quality dimensions onto discrete scores and verifiable rubrics. Improving judge performance will require addressing these competency gaps rather than merely refining evaluation protocols.

Table 7: Performance comparison between single LLM and the agent-integrated ensemble.

| Model | Agreement |
|---|---|
| gpt-4.1 | 65.0 |
|     +UI-TARS-1.5 | 66.2 |
| DeepSeek-V3-0324 | 66.3 |
|     +UI-TARS-1.5 | 67.6 |

## 5    CONCLUSION

In this work, we introduce WEBDEVJUDGE, a comprehensive benchmark for evaluating LLM-as-a-judge in web development. Unlike previous benchmarks, WEBDEVJUDGE supports both static code analysis and interactive agent navigation with high-quality preference labels. Our experiments demonstrate that current LLM-as-a-judge approaches cannot effectively substitute human evaluation, and we identify core bottlenecks hindering their performance. Since our primary focus is on evaluation and analysis within the general LLM-as-a-judge domain, we did not specifically optimize the overall framework structure. We leave the exploration of sophisticated agentic workflows and complex multi-round evaluations to future work.

### ACKNOWLEDGMENTS

Some authors of this paper were supported by the ITSP Platform Research Project (ITS/189/23FP) from ITC of Hong Kong, SAR, China, and the AoE (AoE/E-601/24-N), and the GRF (16205322) from RGC of Hong Kong, SAR, China. We also thank the support from Tencent AI Lab.

### ETHICS STATEMENT

The ethical considerations of our work are discussed in the context of the following aspects: (1) **Data collection and use**. We use publicly available datasets and self-generated data for evaluation. We ensure that the data is only used for academic research purposes and no personal data is involved. We strictly follow the license terms and conditions. (2) **LLMs API**. We comply with the terms of service of the LLMs API providers strictly, maintaining fair use.

### REPRODUCIBILITY STATEMENT

We provide detailed descriptions of our methods, datasets, and evaluation metrics in the main text and appendix to ensure transparency and reproducibility. Our benchmark construction process for WEBDEVJUDGE is detailed in Section 3 and Appendix B. The experimental setup, including the models, evaluation paradigms, and specific protocols, is described in Section 4 and Appendix C.

We also provide a complete description of the *WebDevJudge-Unit* dataset in Appendix D. Code and scripts are provided in the supplementary materials to replicate the empirical results.

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

## A    LIMITATIONS OF WEBDEVJUDGE

While WEBDEVJUDGE establishes a systematic framework for benchmarking LLM preference pre-diction, we recognize several constraints that reflect the inherent complexity of modeling human judgment. The core of preference modeling lies in its inherent subjectivity. Although we provided annotators with comprehensive guidelines to standardize evaluations, personal preferences are difficult to fully decouple from objective quality. Because our annotation pool is limited in size, the resulting labels may reflect the collective biases of a specific demographic or professional group rather than a universal human consensus, especially for ambiguous cases. The current scale of WEBDEVJUDGE is relatively focused. However, we have retained the broader set of filtered samples, providing a pathway for future data augmentation.

## B    DETAILS IN WEBDEVJUDGE CONSTRUCTION

### B.1    DATA COLLECTION AND FILTERING PIPELINE

Our data collection process commences with the `webdev-arena-preference-10k` dataset (Vichare et al., 2025; Chiang et al., 2024), which contains 10,501 user queries, each paired with two web implementations and a user-provided preference label. To ensure the quality and suitability of this data for our benchmark, we implemented a rigorous two-stage filtering pipeline. The overview of the filtering pipeline can be seen in Table 8.

Table 8: Overview of the filtering pipeline, including the number of instances before and after filtering, and the purpose of each filtering stage.

| Stage and criterion | # before | # after | Purpose |
|---|---|---|---|
| Query-based: verbatim-identical duplicate | 10,501 | 6,730 | Remove redundancy while maintaining the original distribution. |
| Query-based: intention, interaction and safety | 6,730 | 2,460 | Remove harmful, offensive, or nonsensical content, and queries with minimal interaction requirements and unclear intentions. |
| Env-based: deployment failure checking via screenshot | 2,460 | 1,814 | Remove instances with deployment failures. |
| Env-based: deployment failure checking via status code | 1,814 | 1,713 | Remove instances with deployment failures. |
| Sampling | 1,713 | 700 | Sample 700 instances regarding the cost-effectiveness. |
| Manual filtering during annotation | 700 | 654 | Filter out instances with harmful content and deployment failures. |

**Query-based Filtering**    The raw queries were first processed to address quality issues. We began by removing all verbatim duplicate queries to eliminate redundancy. Subsequently, we employed `gemini-2.5-pro` to screen for and exclude any queries containing harmful, offensive, or nonsensical content, based on a predefined set of safety and clarity instructions. We then exclude queries with minimal interaction requirements (score $< 8$, max $= 10$) and queries with unclear intentions (score $< 3$, max $= 5$). The prompts used for filter stages are presented below.

**Environment-based Filtering**    Following the query-based filtering, we proceeded to validate the web implementations. We established a standardized Next.js environment for deployment. To manage dependencies, we identified and included the most common packages, thereby excluding implementations that required niche or incompatible libraries. Each successfully deployed implementation was then verified via a test request; those returning a non-200 status code were discarded as buggy. Finally, to handle instances with intrinsic runtime errors not captured by status codes (e.g., rendering a blank page), we captured an initial screenshot of each webpage. A multimodal

model, `Qwen2.5-VL-72B`, was utilized to visually inspect these screenshots and filter out any pages exhibiting rendering failures.

This comprehensive filtering process yielded a final set of 1,713 high-quality instances for subsequent annotation. We further sampled 700 instances from the final set for annotation. During annotation, we filtered out instances with harmful content and deployment failures manually. While the original dataset includes user-provided preference labels, we identified several factors that render them unreliable for rigorous evaluation. Primarily, the labels are susceptible to a high degree of subjectivity and variance in evaluation criteria among individual users. The single-pass nature of the crowdsourced annotations also lacks the verification necessary for robust benchmark data. Furthermore, the dataset includes a "tie (bothbad)" category, which introduces ambiguity. The subjective definition of "bad" can lead to inconsistent labeling, potentially masking instances where one implementation, though imperfect, is demonstrably superior to the other. To illustrate these inconsistencies, we provide examples of problematic cases from the original dataset in Table 9.

---

**PROMPT FOR QUERY-BASED INTERACTION FILTERING**

Please analyze the user's input to assess the level of detail regarding interaction requirements for web design. Focus on the functionality implied by the user's input and the interactions required. Based on your evaluation, assign a score from 0 to 10:

0: No potential interaction requirements are mentioned, but the website may still contain interactions.
3: No interaction is mentioned explicitly, but interactions may still exist in the functionality.
5: One interaction is mentioned or implied, focusing on a single functionality or task.
8: More than one interaction is mentioned or implied, involving multiple functionalities or tasks.
10: All relevant interactions and functionalities are clearly mentioned in detail, requiring full user engagement with the site or app.

After completing your analysis, output your score in the format: `<<score:int>>`. For example: `<<score:7>>`.

Examples:

User: Building a merge sort function Response: This is a programming task that doesn't involve any web interactions or user-facing functionality. `<<score:0>>`

User: a poker site Response: The term "poker site" implies interactions, but the input lacks specific details about those interactions. `<<score:2>>`

User: Create a unique and incredibly beautiful design. The design is up to your discretion, but the main thing is to make it stand out! Write code that is at the senior level or even above it, as if you've been working in the frontend for 10 years. Make the design beautiful and pleasing to the eye! Response: The focus is on aesthetics and quality, with some implied user interaction related to the design's appeal, but not much about functionality. `<<score:3>>`

User: site for selling craft chocolate bars Response: This request implies basic interactions such as browsing and purchasing, but lacks specific details about the interactions. `<<score:5>>`

User: Design a guided meditation player with progress tracking. Create a library view to browse and select different meditation sessions. Response: The request involves multiple interactions like browsing, selecting sessions, and tracking progress, but is not fully detailed. `<<score:8>>`

User: Build a game of chess Response: A chess game involves numerous interactions such as moving pieces, taking turns, and tracking the game state, all requiring user engagement. `<<score:9>>`

User: Make me a clone of WhatsApp Chat App Response: A clone of WhatsApp implies multiple interactions, including sending/receiving messages, media sharing, and managing chats, all of which require significant user engagement and functionality. `<<score:9>>`

---

**PROMPT FOR QUERY-BASED SAFETY FILTERING**

You are given a user query. Classify whether it is safe or unsafe.
Definitions:
SAFE: The query is clear, relevant, and does not request or promote hazardous, violent, hateful, or sexually explicit content.
UNSAFE: The query contains harmful, offensive, or nonsensical content.
Output your decision in **valid JSON** with the following schema:

```
{
  "query": "<the original user query>",
  "label": "SAFE" or "UNSAFE",
  "category": "<one of: hazard, violence, sexual, hate, other>",
  "reason": "<brief explanation>"
}
```

---

**PROMPT FOR QUERY-BASED INTENTION FILTERING**

Analyze the user's input to evaluate the clarity of their query regarding website design. Assign a score from 1 to 5 based on the criteria below:
Score 5: User provides a clear, detailed description of requirements, including specific features and design elements.
Score 4: User's intention is mostly clear with some details, but lacks comprehensive specifics.
Score 3: User presents a general idea, but the request is vague and lacks essential information.
Score 2: User's input is unclear or contains irrelevant information, making their intention difficult to discern.
Score 1: Input is nonsensical or purely code-related, reflecting no intention for a design request. Output: Return a JSON object containing the reasoning for the score and an expected result, formatted as follows:

```
[{"reason": "A clone of WhatsApp Chat App", "score": "4"},]
```

---

## B.2 RUBRIC ANNOTATION PIPELINE

The annotation was conducted by two of the authors, all of whom possess strong backgrounds in computer science and software engineering. Since we have two annotators, the agreement rate is calculated as the proportion of instances where their annotations are the same. The annotation requires annotators to interact with the deployed web implementations and provide a preference label via pairwise comparison. However, this direct approach yields low inter-annotator agreement, underscoring the high degree of subjectivity inherent in the task and the need for a standardized evaluation framework.

To address this, we introduced a rubric-guided annotation process. Manually creating detailed rubrics for each query is not only prohibitively time-consuming—with an estimated 20 minutes per rubric, excluding research time—but also intellectually demanding. It requires extensive background knowledge, including familiarity with established UI/UX design patterns for similar applications and, in many cases, specialized domain knowledge for tasks like scientific simulations. Recognizing these challenges, we leveraged a powerful large language model, `gemini-2.5-pro`, for automated generation. Based on each user query, the model produced a structured rubric tree organized along three core dimensions critical to web development quality:

- **Intention**: The core requirements of the user query.
- **Static Quality**: The assessment of static elements, including UI layout and UX design.
- **Dynamic Behavior**: The evaluation of interactive features.

The prompt used to generate these rubric trees is provided below.

Upon manual review, the LLM-generated rubrics exhibited both strengths and weaknesses. For queries that referenced real-world applications, the rubrics were often of high quality, sometimes surpassing human-authored versions in detail. However, for some general queries, the generated rubrics sometimes included criteria that were either too specific or too broad, reflecting the ambiguity of the original request. We present several examples of the rubric tree structure, encompassing the

Table 9: Examples of problematic cases. In each example, the upper figure is referred to as model_a, while the lower one is model_b.

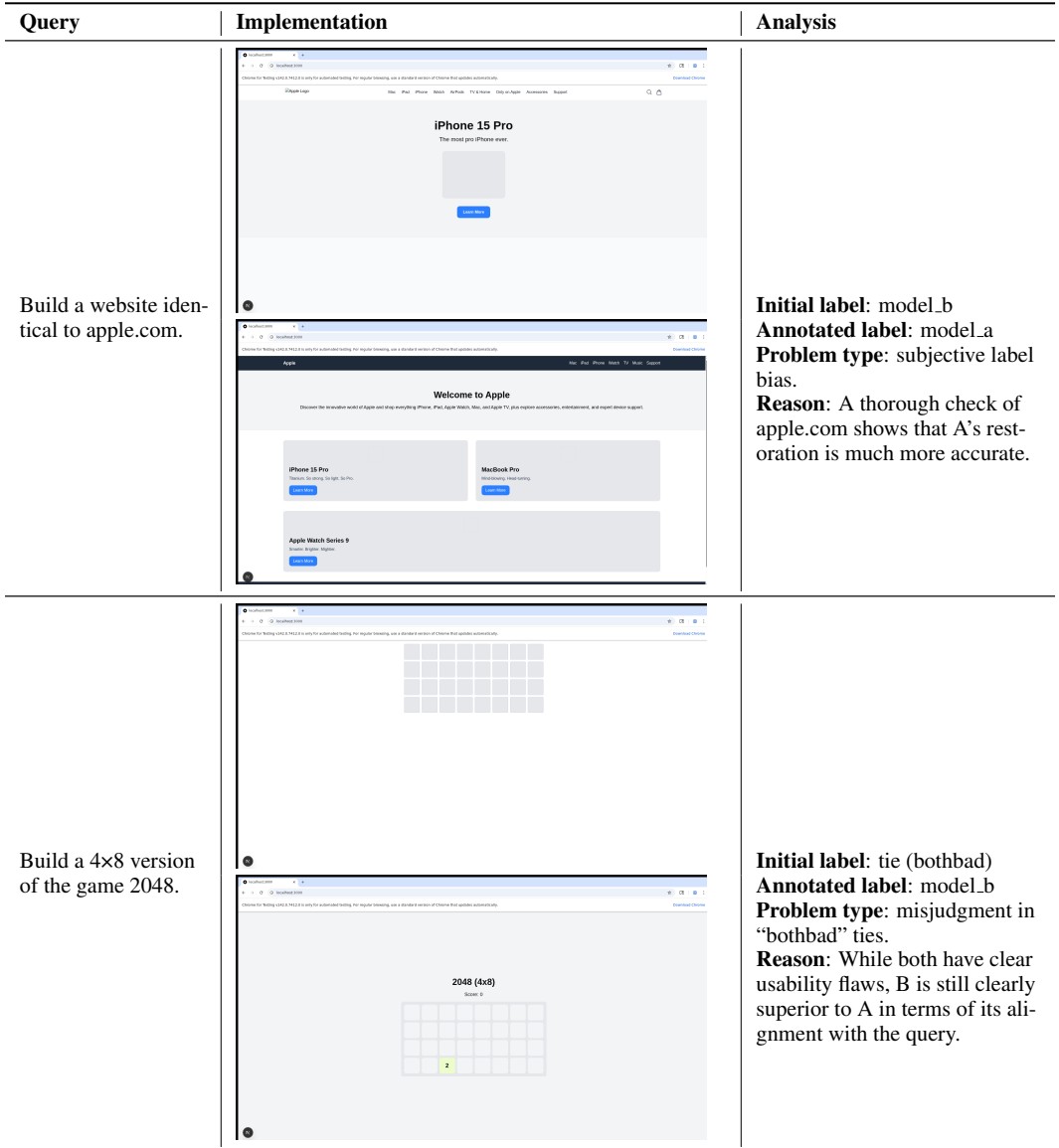

| Query | Implementation | Analysis |
|---|---|---|
| Build a website identical to apple.com. | | **Initial label**: model_b
**Annotated label**: model_a
**Problem type**: subjective label bias.
**Reason**: A thorough check of apple.com shows that A's restoration is much more accurate. |
| Build a 4×8 version of the game 2048. | | **Initial label**: tie (bothbad)
**Annotated label**: model_b
**Problem type**: misjudgment in "bothbad" ties.
**Reason**: While both have clear usability flaws, B is still clearly superior to A in terms of its alignment with the query. |

manually-curated example provided in the one-shot rubric generation, alongside instances of both relatively high and low quality LLM-generated rubrics, as illustrated in Figure 9 in Appendix F.1.

Table 10: Statistics of the generated rubric trees.

| Metric | Intention | Static | Dynamic | Whole |
|---|---|---|---|---|
| Average height | 2.0 | 3.9 | 3.3 | 5.0 |
| Average number of leaf nodes | 3.6 | 15.6 | 10.6 | 29.9 |

We also provide the statistics of the rubric trees; the details are shown in Table 10. From the statistics, especially the average number of leaf nodes, we can see that the LLM-generated rubrics cover a wide range of criteria, which is a good sign for the evaluation of web development tasks. To ensure consistent application of these rubrics, we established a set of clear annotation guidelines for the

expert annotators, as detailed in the following. This structured pipeline, combining LLM-generated rubrics with clear human oversight and guidelines, proved highly effective.

---

**ANNOTATION GUIDELINES**

1. All judgments must be based on actual user experience, not solely on visual appearance.
2. In general, the completeness of functionality takes precedence over aesthetics, unless the user query explicitly requests a focus on design.
3. If both web pages are well-implemented, aesthetic quality should be considered as a deciding factor.
4. If there is any important and discernible difference in quality between the two pages, a preference should be stated rather than defaulting to a tie.
5. When a decision is difficult, the rubric tree should serve as the definitive guide. Refer to the fulfillment of both leaf-level and root-level criteria.
6. When judging based on the rubric tree, consider functional equivalence rather than demanding literal, identical implementations.

---

**PROMPT FOR RUBRIC TREE BUILDING**

## TASK DESCRIPTION
You are an expert software quality assurance (QA) analyst. Your task is to take a user query for a web development project and generate a structured, hierarchical rubric. This rubric will be used to evaluate a generated webpage in a verifiable, binary (implemented/not implemented) manner.
The output must be a single JSON object with ```json and ``` wrapped around it.
The JSON object must have three top-level keys: 'intention', 'static', and 'dynamic'.
### JSON Structure Rules:
1. Each node in the tree must be a dictionary with two keys:
    - 'description': A string describing the feature or goal.
    - 'children': A list of child nodes, or None if it is a leaf node.
2. The 'intention' section should capture the high-level purpose and core goals of the webpage. Descriptions should be concise and conceptual overviews of what the user wants to achieve.
3. The 'static' section must detail all the non-interactive, visible elements of the webpage. Break down components into their smallest logical parts. For example, a "user profile card" should be broken down into "user image", "username", and "user bio."
4. The 'dynamic' section describes all the interactive functionalities of the page.
    - It must have exactly two children: one for "basic" interactions and one for "complex" interactions.
    - 'basic': These are simple, single-step user actions. Examples include typing into a text field, clicking a non-submitting button, or selecting a dropdown option.
    - 'complex': These are multi-step processes or actions that result in a significant change to the application's state. Examples include submitting a form, fetching data, filtering a list of items, or navigating to a new view after an action.
5. Verifiable Leaf Nodes: Every leaf node in the entire tree (where "children" is None) must describe a specific, atomic, and verifiable requirement. The description should be a clear statement that can be evaluated as "implemented" or "not implemented".
## Example:
### User Query:
{example_query}
### Generated Rubric Tree (Your Output):
```json
{example_rubric_tree}
```

Now, analyze the following user query and generate the rubric tree in the specified JSON format.
### User Query:
{user_query}

---

### B.3 DATA STATISTICS

In this section, we describe the categorization pipeline for WEBDEVJUDGE and present representative examples for each subcategory. We adapt topics from the original dataset (Vichare et al., 2025) to serve as our subcategories. Given that WEBDEVJUDGE contains only 654 instances, we did not use a topic modeling model for clustering. Instead, we generated subcategories by providing the

topic name, a detailed description, and the user query to `GPT-4o`. These subcategories were then manually reviewed and consolidated into three main categories.

## C  EXPERIMENTAL DETAILS

In this section, we show the details of the implementation of our experiments. Including the models and hyperparameters, the details of evaluation protocols and metrics, and the agentic workflow implementation.

### C.1  MODELS AND HYPERPARAMETERS

Our experiments utilize a combination of open-source and commercial models. Open-source models were deployed using vLLM (Kwon et al., 2023) and SGLang (Zheng et al., 2024). Commercial models were accessed via their respective APIs, including Azure[1] and Vertex AI[2]. To ensure reproducibility, we set the temperature to 0.0 for all non-reasoning models. For reasoning models, the temperature was set accordingly. For brevity, the models referred to as `DeepSeek-V3` and `DeepSeek-R1` in the main paper correspond to `DeepSeek-V3-0324` and `DeepSeek-R1-0528`, respectively.

### C.2  EVALUATION PROTOCOLS AND METRICS

**Likert Scale**   Motivated by the work of Bian et al. (2025), we designed a multi-level Likert scale for evaluating web development tasks. The scale is based on the international standard for software quality assessment (ISO/IEC/IEEE, 2022) and has been specifically adapted for our dataset. The details of the Likert scale are shown in "Dimensions of the Likert Scale".

In selecting these dimensions, we considered the typical workflow of human evaluators in web testing. Given that the tasks in our dataset are primarily front-end focused, and acknowledging the current limitations of generative models in producing efficient, full-stack solutions, we have concentrated on fundamental aspects of web development, omitting higher-level criteria such as backend performance and efficiency.

For single-implementation evaluation, the model assigns a score from 1 (lowest) to 5 (highest) to each sub-dimension. The final score is the sum of all sub-dimension scores. For pairwise comparison, both implementations are presented to the model simultaneously, which allows it to assign scores based on their relative merits. The final preference is determined by the score difference between the two implementations. A preference is declared for the higher-scoring implementation if the score difference exceeds a threshold of 1; otherwise, the outcome is considered a tie.

**Rubric**   For rubric-based evaluation, the process is guided by the LLM-generated rubric tree. The model is provided with the rubric, the user query, and the web implementation(s). To mitigate the inherent ambiguity in rubric-based assessments, we instruct the model to recognize functional equivalence. For instance, a requirement for a heading is considered met if a heading element is present, even if its text or styling differs from a literal interpretation of the rubric.

In the single-answer grading setting, the model assigns a binary label (pass or fail) to each leaf node of the rubric. We then compute a pass rate of the leaf node for each of the three primary dimensions: intention, static, and dynamic. The final score is the weighted sum of these pass rates. For pairwise comparison, the model evaluates the two implementations against each leaf node, determining which is superior or if they are tied. This yields a win rate for each implementation of the leaf nodes across the three dimensions. A final score for each is calculated as a weighted sum of these win rates. Preference is awarded to the implementation with the higher final score. In our implementation, dimension weights for intention, static, dynamic are set to 1, 3, 2, respectively.

**Direct**   For direct evaluation, instead of providing any criteria, we instruct the model to directly output its preference (i.e., $W_a$, $W_b$, or tie).

---

[1]https://learn.microsoft.com/en-us/azure/cognitive-services/openai/reference
[2]https://cloud.google.com/vertex-ai/generative-ai/docs/learn/overview

---

**DIMENSIONS OF THE LIKERT SCALE**

**Evaluation Criteria**
1. Functional Correctness and Completeness
- 1.1 **Core Functionality**: Evaluates if the primary features and requirements specified in the user query are implemented correctly and function as expected.
- 1.2 **Content Accuracy and Completeness**: Assesses if all the required content (text, images, links, etc.) is present, accurate, and correctly placed as per the user's query.
- 1.3 **Boundary Conditions and Corner Cases**: Examines the solution's behavior with unexpected or extreme user inputs.
- 1.4 **Error Handling**: Evaluates the system's ability to handle errors gracefully. This includes providing clear, user-friendly error messages and preventing application crashes due to invalid operations.

2. User Interface Quality
- 2.1 **Visual Consistency and Cohesion**: Assesses the consistency of design elements such as color schemes, typography, spacing, and component styling throughout the webpage.
- 2.2 **Layout, Structure, and Responsiveness**: Evaluates the overall layout and structural organization of the content. This also critically assesses the responsiveness of the design across different screen sizes (desktop, tablet, mobile).
- 2.3 **Aesthetic Appeal**: Assesses the overall visual appeal of the webpage. This includes the effective use of color, typography, imagery, and whitespace to create an engaging and modern user interface.

3. Code Quality
- 3.1 **Readability and Maintainability**: Assesses the clarity and organization of the code. This includes proper indentation, meaningful variable names, comments where necessary, and a logical file structure.
- 3.2 **Modularity and Reusability**: Evaluates whether the code is broken down into logical, reusable components or functions, avoiding monolithic structures and code duplication.
- 3.3 **Scalability and Efficiency**: Assesses the efficiency of the code, as well as the ability to scale the codebase for future enhancements or new features.

4. Interactivity:
- 4.1 **Effectiveness**: Assesses the functionality and user experience of interactive elements like buttons, forms, menus, and sliders. This includes visual feedback on user actions (e.g., hover states, loading indicators).
- 4.2 **Logical Correctness**: Evaluates whether the application state changes correctly in response to user interactions.
- 4.3 **Accessibility**: Evaluates how easy and intuitive it is for a user to navigate the webpage and interact with its elements to achieve their goals.

## C.3 AGENTIC WORKFLOW IMPLEMENTATION

This section details our agentic workflow implementation, as shown in Table 11. To ensure comparability with our other evaluation methods, the workflow's planner adopts the LLM-generated rubric tree to guide the executor's verification actions. The executor is UI-TARS-1.5 (Seed, 2025), one of the state-of-the-art end-to-end GUI agents operating on a ReAct-style paradigm (Yao et al., 2023), where it first generates a thought and then a corresponding action. These actions are converted into executable pyautogui[3] code. We utilize the official UI-TARS-1.5 action space and have designed specific prompting strategies for each rubric dimension. For the Static dimension, the rubric tree is converted into a list of elements; the agent navigates the page to find these elements and returns a list of those present. For the Dynamic and Intention dimensions, each rubric leaf node becomes a task for the agent to complete. Upon reaching the maximum number of steps, the agent provides a conclusion on the task's outcome. Finally, the summarizer calculates a pass rate for each dimension

---

[3]https://pyautogui.readthedocs.io/en/latest/

based on the agent's findings, and the final score is the weighted sum of these pass rates, mirroring the single-implementation rubric evaluation.

Table 11: Details of the agent settings.

| Dimension | Input Format | Output Format | Max Step |
|---|---|---|---|
| Static | List of elements to check | List of found elements | 6 |
| Dynamic | Task description | Task feasibility and conclusion | 5 for basic, 15 for complex |
| Intention | Intention description | Feasibility conclusion | 15 |

---

**ACTION SPACE OF UI-TARS-1.5**

```
click(point='<point>x1 y1</point>')
left_double(point='<point>x1 y1</point>')
right_single(point='<point>x1 y1</point>')
drag(start_point='<point>x1 y1</point>', end_point='<point>x2 y2</point>')
hotkey(key='ctrl c') # Split keys with a space and use lowercase. Also, do not use
more than 3 keys in one hotkey action.
type(content='xxx') # Use escape characters \', \", and \n in content part to ensure
we can parse the content in normal python string format. If you want to submit your
input, use \n at the end of content.
scroll(point='<point>x1 y1</point>', direction='down or up or right or left') #
Show more information on the 'direction' side.
wait() # Sleep for 5s and take a screenshot to check for any changes.
finished(content='xxx') # Use escape characters \', \", and \n in content part to
ensure we can parse the content in normal python string format.
```

---

## C.4 PROMPT TEMPLATES

The prompt templates used for pairwise comparison experiments are as follows. For single-answer grading, the only action is simply to modify the input presentation, for example, by modifying two separate code blocks into one.

---

**PROMPT FOR DIRECT COMPARISON**

You are tasked with comparing two React code snippets (Model A and Model B) based on the user's query. The input will contain:
- User Query: The question or concern the user has.
- Answer from Model A: The output of Model A.
- Answer from Model B: The output of Model B.
Your output should be in the following JSON format:
```
{
"reason": "Detailed explanation for why one model is better, based on the user's query",
"winner": "'model_a', 'model_b', or 'tie'"
}
``` - reason: A detailed explanation of why one model's answer is better than the other, based on the user's query. Focus on the quality of the responses and how well they address the user's concerns.
- winner: Select the winner from the following options:
- "model_a": If Model A is better.
- "model_b": If Model B is better.
- "tie": If both models are equally good or both provide unsatisfactory answers.
Ensure that you thoroughly evaluate the responses before selecting the winner and providing the reason.

---

**PROMPT FOR RUBRIC-BASED COMPARISON**

You are an expert Quality Assurance engineer specializing in web development. Your objective is to meticulously evaluate and compare two different web development solutions for the same task based on a predefined rubric. You will be provided with the user's initial query, two solutions (codes for both webpages A and B), and a comprehensive rubric covering intention, static, and dynamic elements of the webpage.

Based on these inputs, you will assess whether each requirement in the rubric is implemented in each of the two solutions. Avoid any position biases and ensure that the order in which the responses were presented does not influence your decision. Do not allow the length of the responses to influence your evaluation. Be as objective as possible. During your assessment, please note that the solution might use different terminology than the rubric. Consider a requirement met if the solution's feature is equivalent. For example, the required heading element is present on the webpage, though the exact text or symbol differs.

## User Query
{user_query}

## Code A
```tsx
{code_a}
```

## Code B
```tsx
{code_b}
```

## Rubric
### Intention
```json
{intention_rubric} ```
### Static Elements
```json
{static_rubric}
```

### Dynamic Elements
```json
{dynamic_rubric}
```

## INSTRUCTIONS Your task is to return a single JSON object. This object should have three top-level keys: "intention", "static", and "dynamic". The value for each key should be a JSON object that mirrors the structure of the corresponding rubric provided above. For each leaf node in each rubric (i.e., where "children" is null), you must add a new key "value". The value for this key must be a string: "A" if solution A is better, "B" if solution B is better, or "tie" if they are of equal quality or both fail to meet the requirement.

## Output Format
Begin your evaluation by providing an explanation for your reasoning. End your output with a JSON object wrapped with ```json at the beginning and ``` at the end. Do not include any other text after the JSON object.

Here is an example of the output format:
```json
{ "intention": { "description": "The purpose of the web page.", "children": [ { "description": "A web page for book reviews.", "children": null, "value": "A" } ] }, "static": { "description": "The static elements of the web page.", "children": [ { "description": "The book review submission form.", "children": [ { "description": "A field to input the book's rating.", "children": null, "value": "tie" } ] } ] }, "dynamic": { "description": "The interaction between the user and the web page.", "children": [ { "description": "Basic user interactions.", "children": [ { "description": "User can type text into the review text area.", "children": null, "value": "B" } ] } ] } } ``` """
```

---

**PROMPT FOR LIKERT SCALE-BASED COMPARISON**

You are an expert Quality Assurance engineer specializing in web development. Your objective is to meticulously evaluate and compare two different web development solutions for the same task. You will be provided with the user's initial query and the solutions (codes for both webpages A and B). Based on these inputs, you will assess the quality of each solution across several key dimensions. For each sub-criteria, you must provide a rating on a 5-point Likert scale, where 1 represents "Very Poor" and 5 represents "Excellent". Avoid any position biases and ensure that the order in which the responses were presented does not influence your decision. Do not allow the length of the responses to influence your evaluation. Be as objective as possible.

**{DIMENSIONS OF THE LIKERT SCALE}**

## User Query
{user_query}

## Code A
```tsx
{code_a}
```

## Code B
```tsx
{code_b} ```

## Output Format
Begin your evaluation by providing a short explanation. End your output with a json object wrapped with ```json at the beginning and ``` at the end. Use the sub-criterion id as the key. The value for each key should be a nested json object containing the scores for each solution, with "A" and "B" as keys. Do not include any other text after the json object.
Here is an example of the output format:
```json
{ "1.1": { "A": 5, "B": 4 }, "1.2": { "A": 4, "B": 5 }, "1.3": { "A": 3, "B": 3 }, "1.4": { "A": 2, "B": 2 }, "2.1": { "A": 5, "B": 5 }, "2.2": { "A": 4, "B": 3 }, "2.3": { "A": 3, "B": 4 }, "3.1": { "A": 5, "B": 5 }, "3.2": { "A": 4, "B": 4 }, "3.3": { "A": 3, "B": 3 }, "4.1": { "A": 5, "B": 4 }, "4.2": { "A": 4, "B": 5 }, "4.3": { "A": 3, "B": 4 } }
```

## D WEBDEVJUDGE-UNIT DATASET

In this section, we introduce WebDevJudge-Unit, a task-level dataset created to assess the capability of evaluators to verify task feasibility.

### D.1 DATASET CONSTRUCTION

To construct the WebDevJudge-Unit dataset, we began by randomly sampling 105 queries from WEBDEVJUDGE. For each query, we prompted `gemini-2.5-pro` to generate up to five corresponding verification tasks, each with an expected result. We then generated the necessary HTML code for each query to facilitate easy deployment. Following deployment, each task was meticulously annotated for feasibility (true/false) by interacting with the live web application. For tasks identified as infeasible, we further annotated the specific error type and provided a detailed reason for the failure.

Table 12: Statistics of the WebDevJudge-Unit dataset.

| Error Type | Proportion | Example | |
|---|---|---|---|
| | | *Description* | *Error Reason* |
| Non-functional element | 35.9 | **Task**: On a Question Review page, modify the text content of a displayed question within its editable text field. **Expected Result**: The displayed question text updates to reflect the new input. | Unable to edit question content. |
| Missing element | 30.9 | **Task**: Click the 'Market' tab. **Expected Result**: The 'Market' view is displayed, showing a list of cryptocurrencies. | Can not find the Market tab. |
| Prerequisite not met | 15.2 | **Task**: Select an answer for a multiple-choice question by clicking on one of the options. **Expected Result**: The chosen answer option is visually highlighted. | Unable to import the quiz correctly. |
| Loading issue | 9.0 | **Task**: Click a file name in the File Explorer sidebar. **Expected Result**: The corresponding file's mock content is shown in an editor tab. | Unable to load file content. |
| Unreasonable outcome | 5.4 | **Task**: Scroll the chat display area to view older messages when content overflows. **Expected Result**: The user is logged out. | N/A |
| Ambiguous input | 2.2 | **Task**: Hover over a data point or segment in a Chart. **Expected Result**: A tooltip containing mock data relevant to the hovered chart element is displayed. | Chart is not required in the webpage. |
| Overly detailed task | 0.9 | **Task**: From a list of activities on the 'Activities Screen', click on a specific activity's name or image. **Expected Result**: The 'Detail View' for the selected activity is displayed, showing its full description. | Unable to get the expected result based on the operation. |
| Missing animation | 0.5 | **Task**: Move the mouse cursor across the main 3D animated background area. **Expected Result**: The 3D background animation dynamically responds to the cursor's position or movement. | The background 3D animation is not responsive. |

### D.2 DATASET STATISTICS AND EXAMPLES

The resulting dataset comprises 502 tasks derived from 105 unique queries. The feasibility labels are distributed as follows: 279 tasks are marked as feasible and 223 as infeasible. Table 12 presents a detailed breakdown of the error types for the infeasible tasks, along with illustrative examples for each category.

# E ADDITIONAL RESULTS

This section provides a more detailed analysis of LLM-as-a-judge performance on web development tasks, supported by supplementary experimental results.

## E.1 MITIGATING THE IMPACT OF POSITIONAL BIAS

To mitigate the impact of positional bias, we employ a widely-used debiasing technique from prior works (Zeng et al., 2024; Tan et al., 2025). This method involves evaluating each pair of implementations twice, with their positions swapped in the second evaluation. A preference is considered final only if the model's choice remains consistent across both orderings. If the choice is inconsistent, the outcome is recorded as a tie. We use this approach to investigate whether mitigating positional bias improves overall model performance. The results are shown in Table 13.

Table 13: Agreement rate (%) with and without mitigating the positional bias.

| Model | w/ mitigating | | w/o mitigating | |
|---|---|---|---|---|
| | *Direct* | *Likert* | *Direct* | *Likert* |
| Claude-4-sonnet | 68.6 | 68.3 | 70.0 | 70.2 |
| GPT-4.1 | 68.8 | 68.3 | 68.7 | 70.3 |
| DeepSeekV3-0324 | 64.2 | 64.4 | 66.7 | 67.1 |

A natural question arises as to why we did not employ the swap-based debiasing technique in our main experiments. Our decision to rely on single-pass evaluation stems from two primary considerations. First, our core objective is to characterize the raw, unfiltered behavior of LLMs-as-judges to understand their inherent biases. Applying debiasing techniques from the outset would mask or average out these effects, obscuring the very phenomena we seek to analyze. The results in Table 13 indicate that the overall performance difference before and after debiasing is not substantial in our setup. This finding reinforces the importance of studying the biases directly, as their presence is not always evident from aggregate performance metrics alone. Second, the single-pass evaluation mirrors many practical application scenarios where, for reasons of cost and efficiency, models are queried only once per pair. Therefore, our main experimental results offer a more realistic and cautionary benchmark for practitioners, highlighting the potential pitfalls of naively deploying these models without safeguards. Our methodology thus deliberately separates the diagnosis of inherent model behaviors from the evaluation of mitigation strategies.

## E.2 PREDICTION DISTRIBUTION CONSISTENCY

To understand the nature of model failures, we analyze the consistency of predictions across different evaluators. This investigation seeks to determine whether different models tend to err on the same set of instances—suggesting certain examples are inherently challenging—or if their errors are largely independent and model-specific. The results of this label consistency analysis are presented in Figure 5 and Figure 6.

Our analysis of prediction consistency across different models reveals a significant disparity between the two evaluation paradigms. As illustrated in Figure 5 and Figure 6, the inter-model agreement under the pairwise comparison paradigm is substantially higher than that under single-answer grading. In pairwise comparisons, the consistency rates between different evaluators generally exceed 75%, with many model pairs even surpassing 80%. In stark contrast, under single-answer grading, inter-model consistency drops significantly, typically hovering between 50% and 65%.

This discrepancy corroborates our core finding from the main text: pairwise comparison is a more stable and reliable paradigm for complex, open-ended tasks like web development. Relative judgment in pairwise comparison constrains the evaluation scope, compelling the model to focus on discriminative features between two candidates, which is a cognitively less demanding task than absolute scoring. Conversely, single-answer grading requires the model to possess a well-calibrated and consistent internal standard of quality, a standard that varies greatly across different models. Consequently, the single-answer grading paradigm exposes the current models' deficiencies in calibration, leading to inconsistent and less reliable evaluation outcomes.

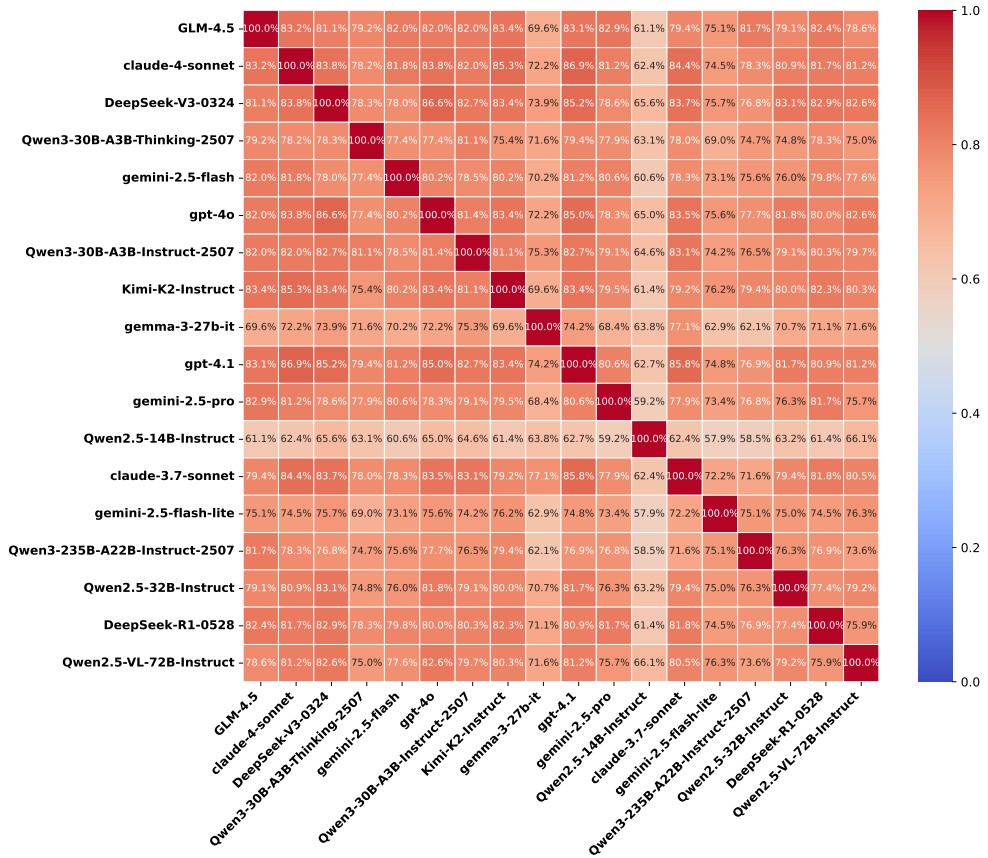

Figure 5: The consistency between model predictions under pairwise comparison.

Furthermore, the high consistency in pairwise comparison suggests that while different models may lack a shared understanding of 'absolute quality,' their internal mechanisms for making 'relative preference' judgments are more aligned. This reinforces our view that leveraging relative judgments is a more robust approach for automated evaluation in domains characterized by nuance and ambiguity, where clear-cut right or wrong answers are scarce.

To further substantiate these findings, we extend our consistency analysis to rubric-based grading, examining its internal consistency across different models (Figure 7) and its alignment with the direct evaluation paradigm (Figure 8).

### E.3 EVALUATION RESULTS ON NON-TIE CASES

Given the brittleness of language models when handling ambiguous tie conditions, we also present results for non-tie cases, where a clear preference was established. As presented in Table 14, excluding tie cases results in a substantial improvement in agreement rates for all evaluators, indicating that ambiguous comparisons are a primary source of error. More importantly, these results reinforce the core conclusions drawn from our main experiments. The pairwise comparison paradigm consistently and significantly outperforms single-answer grading, highlighting its robustness for capturing relative quality. Furthermore, the patterns of different guidance mechanisms remain consistent: in the pairwise setting, both Likert scale and Rubric-based guidance yield comparable performance, suggesting that the relative judgment itself is the dominant factor. In contrast, for single-answer grading, the Rubric-based approach is demonstrably superior to the Likert scale. This reinforces our finding that structured, binary assessments provide a more reliable signal for absolute evaluation than multi-point scales, which require a level of calibration that models currently lack.

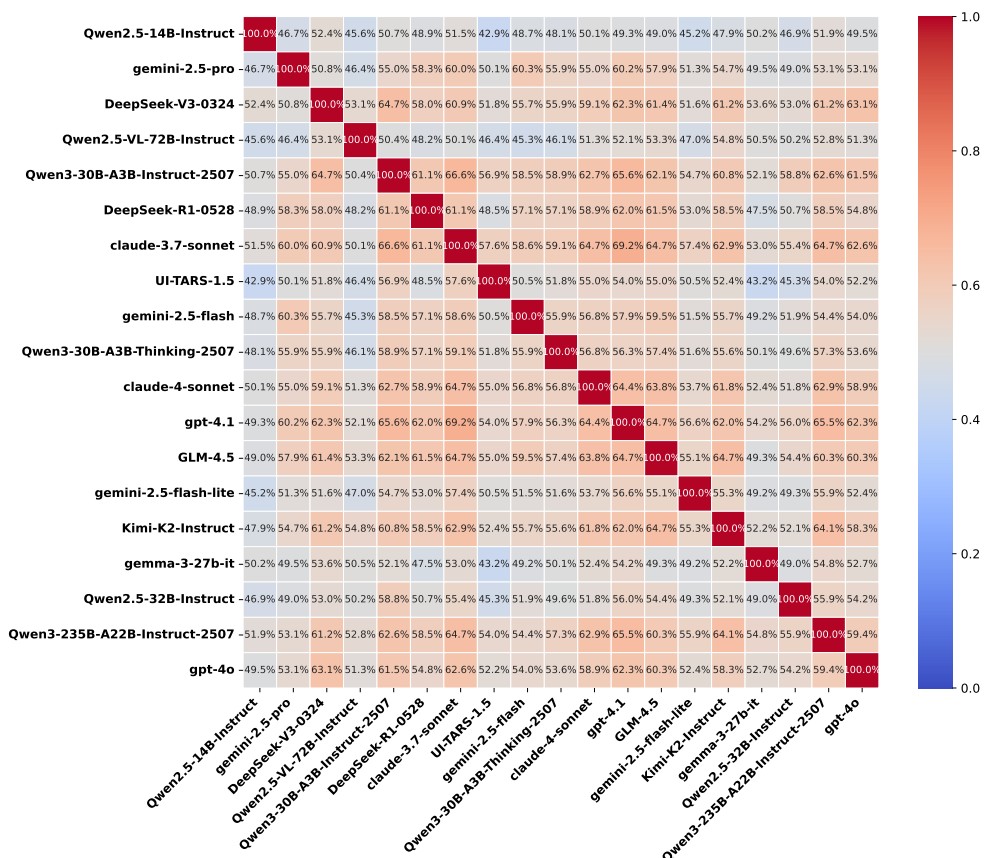

Figure 6: The consistency between model prediction under single answer grading.

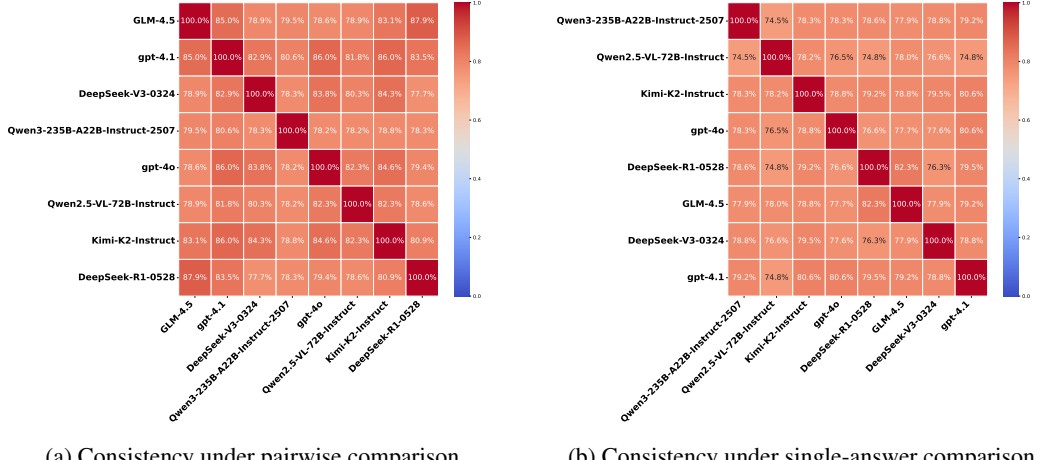

(a) Consistency under pairwise comparison.

(b) Consistency under single-answer comparison.

Figure 7: The inner prediction consistency between model predictions under rubric paradigm.

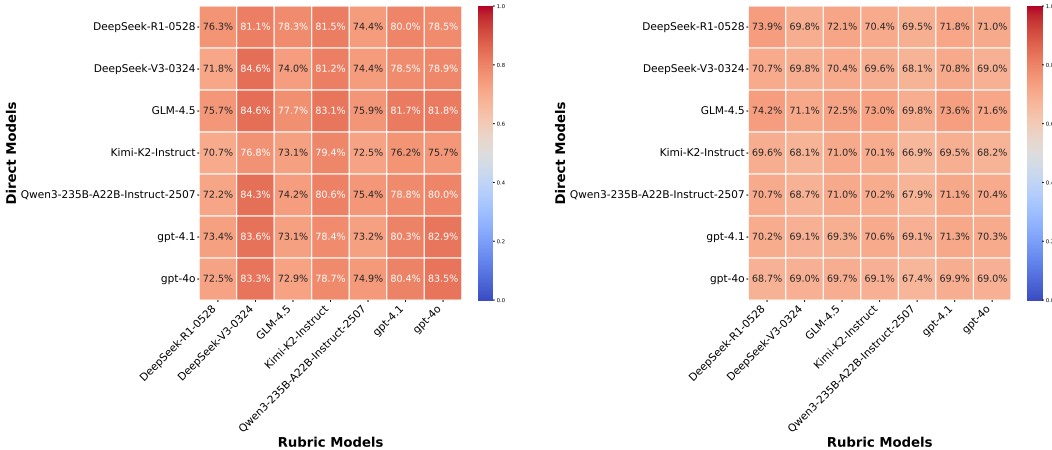

(a) Prediction consistency under rubric (pairwise comparison) and direct paradigm.

(b) Prediction consistency under rubric(single-answer grading) and direct paradigm.

Figure 8: The inter-method consistency between model predictions under rubric paradigm and direct paradigm.

Table 14: Agreement Rate (%) (without tie) of different evaluators under different evaluation paradigms.

| Model/Method | DIGITAL DESIGN | | GAME & APP | | WEB & SPECIAL | | AVERAGE | |
|---|---|---|---|---|---|---|---|---|
| | Single | Pair | Single | Pair | Single | Pair | Single | Pair |
| **Likert** | | | | | | | | |
| 🖼 GPT-4.1 | 67.39 | 83.7 | 72.16 | 84.02 | 69.46 | 82.63 | 69.72 | 83.49 |
| 🖼 GPT-4o | 58.7 | 80.98 | 64.95 | 81.96 | 68.26 | 77.84 | 63.85 | 80.37 |
| 🖼 Qwen-2.5-VL-72B-Inst. | 49.46 | 76.09 | 53.61 | 76.29 | 57.49 | 76.65 | 53.39 | 76.33 |
| 🖼 Gemini-2.5-flash-lite | 57.07 | 70.65 | 57.73 | 70.62 | 55.69 | 70.66 | 56.88 | 70.64 |
| DeepSeek-V3-0324 | 67.93 | 79.35 | 65.98 | 76.8 | 68.86 | 80.84 | 67.52 | 78.9 |
| Kimi-K2-Inst. | 69.02 | 80.98 | 69.07 | 76.29 | 62.87 | 77.84 | 67.16 | 78.35 |
| Qwen3-235B-A22B-Inst. | 63.59 | 77.17 | 69.07 | 78.87 | 66.47 | 73.65 | 66.42 | 76.7 |
| Qwen3-30B-A3B-Inst. | 67.93 | 79.35 | 70.62 | 74.74 | 66.47 | 79.04 | 68.44 | 77.61 |
| Qwen2.5-32B-Inst. | 55.43 | 73.37 | 56.7 | 76.8 | 57.49 | 75.45 | 56.51 | 75.23 |
| Gemma-3-27B-it | 50.54 | 70.65 | 58.25 | 64.95 | 48.5 | 73.05 | 52.66 | 69.36 |
| Qwen2.5-14B-Inst. | 51.63 | 59.78 | 51.03 | 57.73 | 42.51 | 58.08 | 48.62 | 58.53 |
| 🖼 Claude-4-Sonnet | 63.04 | 84.24 | 70.1 | 82.47 | 69.46 | 79.64 | 67.52 | 82.2 |
| 🖼 Claude-3.7-Sonnet | 76.09 | 82.07 | 71.65 | 76.8 | 68.26 | 83.83 | 72.11 | 80.73 |
| 🖼 Gemini-2.5-pro | 59.78 | 76.63 | 59.28 | 80.41 | 51.5 | 72.46 | 57.06 | 76.7 |
| GLM-4.5 | 67.39 | 82.61 | 65.46 | 78.87 | 62.28 | 77.84 | 65.14 | 79.82 |
| DeepSeek-R1-0528 | 59.24 | 80.98 | 67.53 | 79.38 | 57.49 | 73.65 | 61.65 | 78.17 |
| Qwen3-30B-A3B-Think. | 60.33 | 74.46 | 61.86 | 73.71 | 60.48 | 74.85 | 60.92 | 74.31 |
| **Rubric** | | | | | | | | |
| 🖼 GPT-4.1 | 79.89 | 81.52 | 72.16 | 75.77 | 71.26 | 87.43 | 74.5 | 81.28 |
| 🖼 GPT-4o | 79.35 | 80.43 | 72.68 | 77.32 | 72.46 | 83.23 | 74.86 | 80.18 |
| 🖼 Qwen-2.5-VL-72B-Inst. | 72.28 | 78.8 | 74.23 | 80.41 | 73.65 | 79.64 | 73.39 | 79.63 |
| DeepSeek-V3-0324 | 79.89 | 83.15 | 68.56 | 80.93 | 77.25 | 86.23 | 75.05 | 83.3 |
| Kimi-K2-Inst. | 74.46 | 83.15 | 70.1 | 78.35 | 75.45 | 83.83 | 73.21 | 81.65 |
| Qwen3-235B-A22B-Inst. | 78.8 | 78.26 | 69.59 | 69.07 | 71.86 | 74.85 | 73.39 | 73.94 |
| DeepSeek-R1-0528 | 78.8 | 81.52 | 67.53 | 72.16 | 73.65 | 77.25 | 73.21 | 76.88 |
| GLM-4.5 | 79.35 | 82.07 | 69.07 | 73.71 | 77.25 | 79.64 | 75.05 | 78.35 |
| Agentic workflow | 70.11 | - | 67.01 | - | 76.05 | - | 70.83 | - |

# F    CASE STUDY

## F.1    EXAMPLES OF RUBRIC TREES

Examples of the rubric tree structure are illustrated in Figure 9. We can observe that for queries requiring specific domain knowledge (e.g., the mid one), large language models can generate detailed and accurate rubrics, such as evaluation criteria for specific components. This highlights a key advantage of LLMs over humans: the ability to leverage their extensive domain knowledge to generate precise rubrics for corresponding queries. However, for more general queries (e.g., the rightmost query), the rubrics generated by LLMs can be overly fine-grained. This may introduce ambiguity during evaluation, for instance, by adding overly specific categories like "Content" for a general "categories".

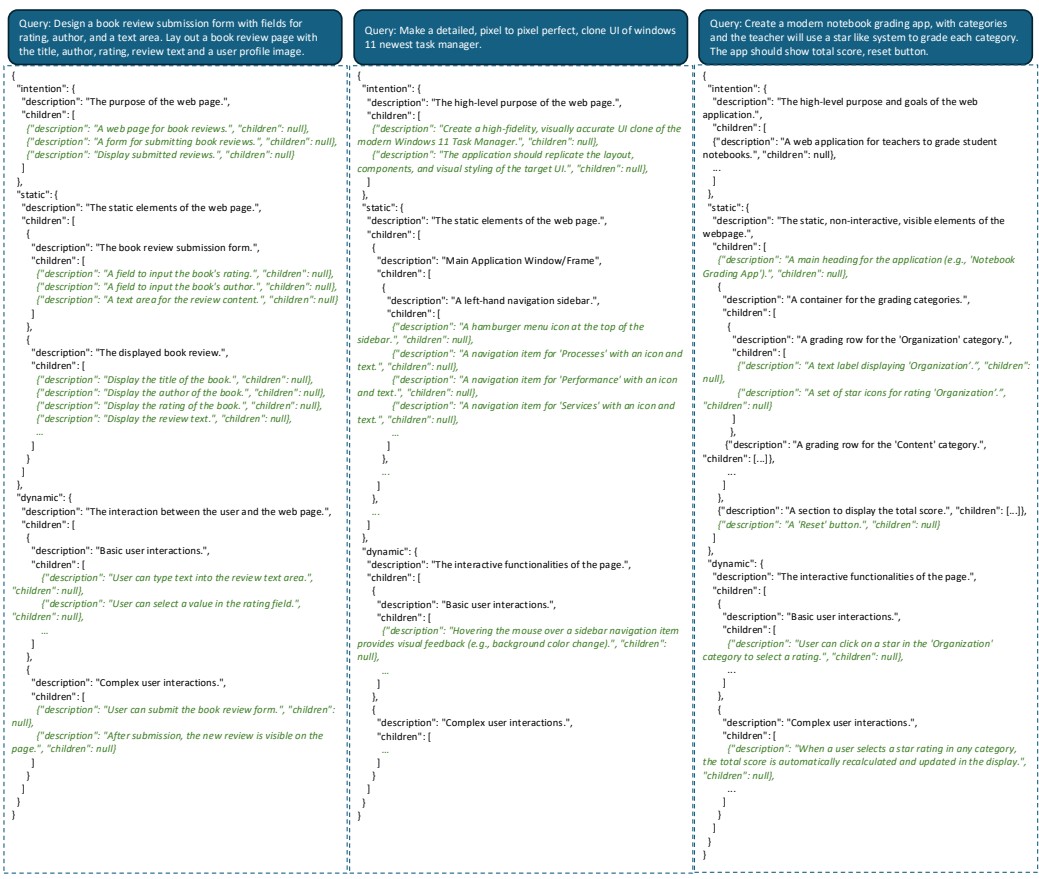

Figure 9: Examples of the rubric tree structure derived from different sources with corresponding queries. The diagram illustrates human-written rubric trees used for few-shot generation (left), alongside good (center) and suboptimal (right) examples of LLM-generated rubric trees. Note that the children attribute of leaf nodes is explicitly set to null. For conciseness, some internal nodes have been omitted. Leaf nodes are labeled in green.

## F.2    FAILURE CASES OF LLM-BASED AND AGENTIC EVALUATORS

In this section, we present failure cases that illustrate the different failure modes discussed in Section 4.

Regarding limitations in understanding functional equivalence, we examine the query and rubric shown in Figure 9 (right). Although the rubric presents some ambiguity, human annotators and GPT-4.1 successfully identify functional equivalence for both the specific category "Organization"

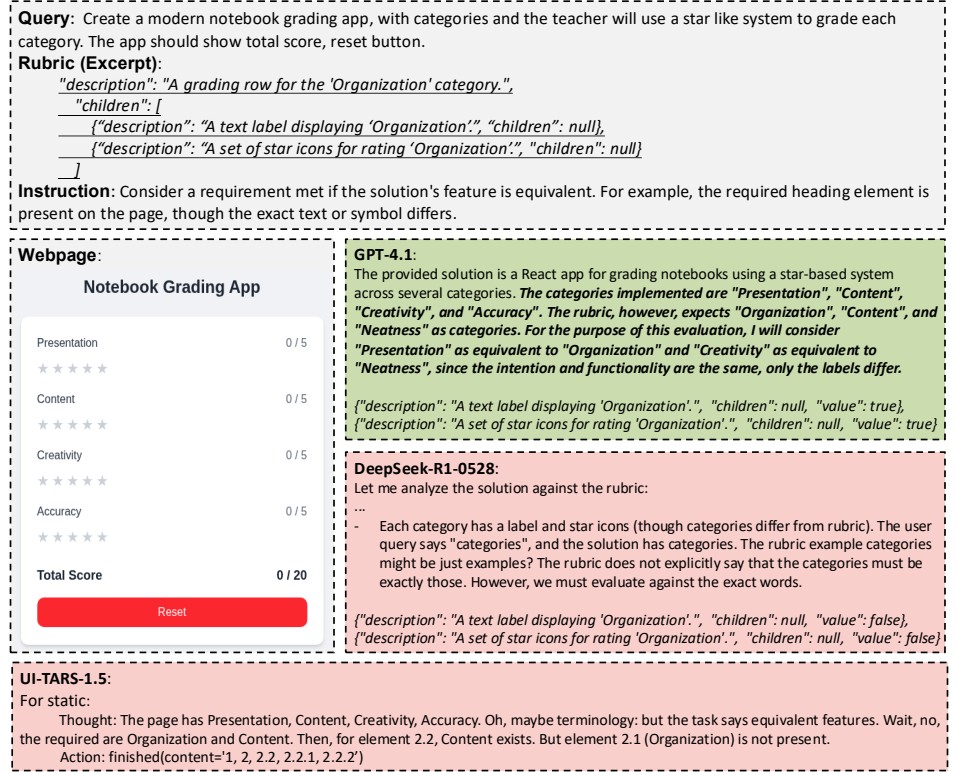

Figure 10: Failure cases in functional equivalence identification. While GPT-4.1 correctly identifies equivalence for the specific "Organization" category and the general "categories" requirement given the query, rubric, and instructions, other models like DeepSeek-R1-0528 and UI-TARS-1.5 fail to do so. These models misinterpret the requirements and exhibit hesitation in recognizing the equivalence for general categories.

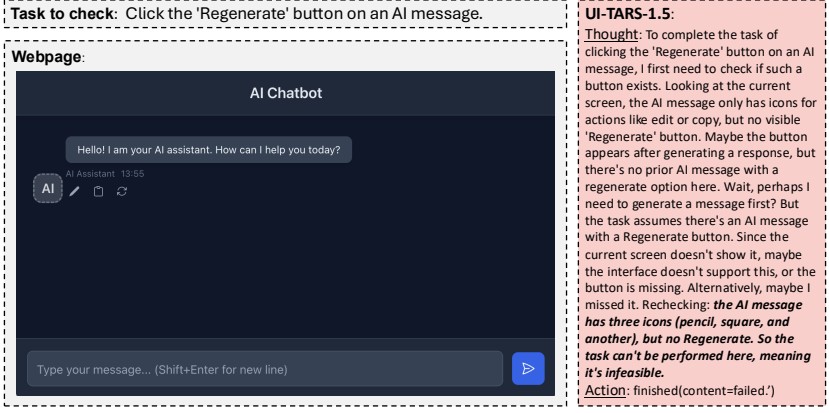

Figure 11: Failure case in feasibility analysis due to operation error. The agentic evaluator fails to detect the target 'regenerate' element in the provided screenshot, resulting in an incorrect assessment of the task as infeasible.

and the general "categories" requirement, provided with the query, rubric, and instructions. In contrast, other models, such as DeepSeek-R1-0528 and UI-TARS-1.5, struggle with interpretation. As shown in Figure 10, these models suffer from misinterpretation, which leads to incorrect functional equivalence identification.

Regarding limitations in feasibility analysis, agentic evaluators are constrained by their inherent capabilities. Figure 11 illustrates such a case where the agent fails to locate the target 'regenerate' element within the screenshot, incorrectly leading it to classify the task as infeasible.

## G   THE USE OF LARGE LANGUAGE MODELS

We use a large language model (LLM) as a general-purpose writing assistant. The primary use of the LLM is to refine and improve the clarity and flow of the text. Specifically, we provided the LLM with our pre-existing draft text and a clear outline of our ideas. We instruct the model to adhere strictly to the provided content, without adding or removing any information. The LLM's role is limited to enhancing the logical coherence and fluency of the language, ensuring the final text is more polished and easier to read.

